# An Age Effect of Rumen Microbiome in Dairy Buffaloes Revealed by Metagenomics

**DOI:** 10.3390/microorganisms10081491

**Published:** 2022-07-25

**Authors:** Long-Ping Li, Ke-Lan Peng, Ming-Yuan Xue, Sen-Lin Zhu, Jian-Xin Liu, Hui-Zeng Sun

**Affiliations:** 1Ministry of Education Key Laboratory of Molecular Animal Nutrition, Zhejiang University, Hangzhou 310058, China; llp_315@163.com (L.-P.L.); pengkelan@zju.edu.cn (K.-L.P.); myxue@zju.edu.cn (M.-Y.X.); 12017016@zju.edu.cn (S.-L.Z.); liujx@zju.edu.cn (J.-X.L.); 2Shaanxi Provincial Engineering and Technology Research Center of Cashmere Goats, Yulin University, Yulin 719000, China

**Keywords:** dairy buffaloes, age, rumen microbiota, metagenomics

## Abstract

Age is an important factor in shaping the gut microbiome. However, the age effect on the rumen microbial community for dairy buffaloes remains less explored. Using metagenomics, we examined the microbial composition and functions of rumen microbiota in dairy Murrah buffaloes of different ages: Y (1 year old), M (3–5 years old), E (6–8 years old), and O (>9 years old). We found that Bacteroidetes and Firmicutes were the predominant phyla, with *Prevotella* accounting for the highest abundance at the genus level. The proportion of Bacteroides and Methanobrevibacter significantly increased with age, while the abundance of genus *Lactobacillus* significantly decreased with age (LDA > 3, *p* < 0.05). Most differed COG and KEGG pathways were enriched in Y with carbohydrate metabolism, while older buffaloes enriched more functions of protein metabolism and the processing of replication and repair (LDA > 2, *p* < 0.05). Additionally, the functional contribution analysis revealed that the genera *Prevotella* and *Lactobacillus* of Y with more functions of CAZymes encoded genes of glycoside hydrolases and carbohydrate esterases for their roles of capable of metabolizing starch and sucrose-associated oligosaccharide enzyme, hemicellulase, and cellulase activities than the other three groups (LDA > 2, *p* < 0.05), thus affecting the 1-year-old dairy buffalo rumen carbohydrate metabolism. This study provides comprehensive dairy buffalo rumen metagenome data and assists in manipulating the rumen microbiome for improved dairy buffalo production.

## 1. Introduction

Ruminants have diverse microbes in their rumen responsible for the conversion of plant mass into host absorbable compounds such as microbial proteins and volatile fatty acids (VFAs). Those microbes work together synergistically and contribute substantially to the nutrient utilization, metabolism, growth performance, and health of the host [1]. Many factors, such as diet, age, breed, and geographical range, have direct or indirect effects on rumen microbes [2,3]. The microbiome profile can vary by the developmental stage; for example, the composition of the rumen microbial community in dairy cows changes significantly from birth to the age of 2 years [4]. Age effect on microbial community and the physiological response from the animal host has also been reported in ruminants such as dairy cows [5], calves [6], or goats during preweaning development [7]. Studies of rumen microbial communities in pre-ruminant calves reported the existence of age-related changes in the rumen microbiota [3,4]. In addition, age-related microbial changes were closely associated with inflammation [8] and methane emission [9]. Previous research has also investigated the relationship between the rectal microbiota and the maturation of newborn dairy calves [10,11]. Overall, the rumen microbial composition of adult cattle is more stable than that of newborn ones, while it is poorly understood whether it changes with age from juvenile (12 months old) to old age (>9 years old) in dairy buffaloes.

Compared with other ruminants, buffaloes can more efficiently use low-quality roughage, agricultural, and industrial by-products that contain high levels of ligno-cellulosic materials and are not easy to digest, which makes them adapt to locally available crop residues and exerts stronger disease resistance [12,13,14]. The mechanism of the higher digestive capacity of buffaloes over cattle has not been fully elucidated. However, it is likely that part of the reason for this phenomenon could be explained by the differences in rumen ecology [15]. Previous studies have shown that buffaloes have different rumen ecology than cattle, with a higher population of cellulolytic bacteria (*Ruminococcus albus*), fungal zoospores, and a greater capacity to recycle nitrogen to the rumen [16]. Therefore, buffalo had higher nutrient digestibilities of DM, OM, CP, NDF, and ADF than cattle [17,18]. Furthermore, the buffaloes had a higher liquid outflow rate that could result in a lower number of total rumen ciliate protozoal populations [19]. In addition, Malakar et al. [20] compared fiber degradation in vitro by inoculum with cow’s and buffalo’s ruminal bacteria and fungi and reported that fiber degradation in buffalo was significantly higher than in dairy cows. A recent metagenomic study also reported that buffalo rumen microbiota has stronger fiber degradation and less methane production potential than dairy cow rumen microbiota [21]. Therefore, it is suggested that the dairy buffaloes may have different compositions and functions in rumen microbiota. However, to the best of our knowledge, limited study has focused on identifying and understanding the establishment pattern and function of ruminal microbes in dairy buffaloes using high-throughput sequencing technology. Only a few studies have reported that analyzed the rumen microbiome of buffaloes using metagenomic approaches or 16S rRNA gene amplicon sequencing [21,22,23,24,25,26,27]. However, age-related rumen microbial functions have been rarely studied in buffaloes using a metagenomic approach.

Using the raw data generated in our previous study comparing the rumen microbial resistome between dairy cows and dairy buffaloes [28], we further attempt to understand the establishment pattern of microbes and functions in relation to different ages of dairy buffaloes from juvenile (12 months old) to old age (>9 years old). The results will expand our understanding of the dynamic changes of rumen microbiota and provide novel insights into strategies for manipulating buffalo rumen microbes.

## 2. Materials and Methods

### 2.1. Ethics Statement

All animal procedures were approved by the Animal Care and Use Committee of Zhejiang University (Hangzhou, China) and were in accordance with the university’s guidelines for animal research.

### 2.2. Animals and Sample Collection

Fifteen healthy dairy Murrah buffaloes were divided into four groups based on ages: young (Y, 1 year old, parity = 0, n = 5), middle (M, 3–5 years old, parity = 1, n = 4), eight years old (E, 6–8 years old, parity = 2, n = 3), and old (O, >9 years old, parity ≥ 3, n = 3) (Appendix A). Animals were obtained from the Buffalo Research Institute, Chinese Academy of Agricultural Sciences, and Guangxi Zhuang Nationality Autonomous Region (Nanning, China). The diet of dairy buffaloes was the same as in our previous study [28], which consisted of 60% corn silage, 20.8% corn, 7.4% wheat bran, 3.2% soybean meal, 6.0% cottonseed meal, and micronutrient supplement (96.0% DM, 13.8% CP, 39.1% DNF and 21.1% ADF). The studied buffaloes were reared on a dairy farm located in Nanning City, which belongs to the sub-tropical climate (22°48′ N, 108°22′ E) in China. All the dairy buffaloes were fed at 09:00 a.m. and 17:00 p.m., with free drinking water. In order to avoid inducing stress for buffaloes due to pain or mishandling when sampling, rumen fluid samples were collected from each buffalo by two skilled experimenters using an oral stomach tube (2.6 m long, 17.0 mm in diameter) with negative pressure [29] after 2 h of morning feeding. The first 50–100 mL of rumen fluids in each sample were discarded to remove the potential saliva contamination, and final 15 mL rumen samples were collected from each animal. Four layers of sterile cheesecloth were used to filter collected rumen fluid samples, and filtered rumen fluid was stored at −80 °C until further analysis.

### 2.3. DNA Extraction and High-Throughput Sequencing

Genomic DNA of rumen fluid was extracted using the CTAB method [30] and purified using the QIAquick PCR Purification Kit (QIAGEN, Hilden, Germany) according to the manufacturer’s instructions, and the quality of DNA was evaluated using a NanoDrop 2000 spectrophotometer (NanoDrop Technologies, Wilmington, DE, USA) (Appendix A) and 1% agarose gel (Appendix A). The Covaris M220 (Covaris Inc., Woburn, MA, USA) was used to generate 300 bp DNA fragments. The TrueSeq DNA PCR-Free Library Prep Kits (Illumina, San Diego, CA, USA) were applied to perform metagenome libraries construction according to the manufacturer’s protocol. Metagenomic sequencing (PE150) was performed on an Illumina HiSeq 3000 platform (Illumina, San Diego, CA, USA).

### 2.4. Bioinformatics Analysis

The paired-end Illumina reads were trimmed of adaptors, and low-quality reads (length < 50 bp or with a quality value < 20 or having N bases) were removed by fastp software (version 0.20.0) [31]. Host contaminating sequences (host genomic and adaptor contaminations) were removed by mapping to the Bubalus bubalis genomes using Burrows-Wheeler Alignment (BWA) software (Version 0.7.9a) [32]. After trimming, a set of high-quality and free of host genomic contaminants reads was obtained for further analysis. The software of MEGAHIT (Version 1.1.2) [33] was used for metagenomic contig assembly from single and mixed samples. Open reading frames (ORFs) were predicted with the length greater than 300 bp of the assembled contigs using the MetaGene (https://metagene.de, accessed on 15 February 2020) [34], and a nonredundant gene catalog was obtained using CD-HIT (version 4.6.1) [35] base on 95% identity and 90% coverage. The reads of each sample and assembly results were mapped to the nonredundant gene sets, and gene abundance was estimated using SOAPaligner software (Version 2.22, http://soap.genomics.org.cn/, accessed on 15 February 2020) based on 95% identity [36]. Microbial abundance in the dairy buffalo rumen of Y, M, E, and O groups was analyzed using R software (version 4.1.3) amplicon package.

### 2.5. Taxonomic and Functional Annotation from Rumen Metagenomes

Representative sequences of nonredundant gene catalog were aligned to the NCBI NR database with an e-value cutoff of 1 × 10^−5^ using Diamond (version 0.8.35) [37] for taxonomic annotations. Taxonomic profiles were annotated at domain, phylum, class, order, family, genus, and species levels. The relative abundances of each taxon were calculated and normalized using the proportion of each taxon identified in the same sample. Microbial taxa with a relative abundance >0.1% in at least 50% of buffaloes within each group were used for downstream analysis.

The sequence of the NR gene catalog was mapped to the Kyoto Encyclopedia of Genes and Genomes database (KEGG, https://www.genome.jp/kegg/, accessed on 15 February 2020) using Diamond (Version 0.8.35) [37] with BLASTP (https://blast.ncbi.nlm.nih.gov/Blast.cgi, accessed on 15 February 2020) type with an e-value of 1 × 10^−5^. Pathway annotation was then conducted using KOBAS 2.0 (KEGG Orthology-Based Annotation System) [38] according to the blast results. A cluster of orthologous groups of proteins (COG) annotation for the representative sequences was performed using Diamond (version 0.8.35) [37] against the eggNOG database with an e-value cutoff of 1 × 10^−5^ [39]. Carbohydrate-active enzyme annotation was conducted using HMMscan (Version 3.1b2) (http://hmmer.org/, accessed on 15 February 2020) against the Carbohydrate-Active enZYmes (CAZymes) Database (http://www.cazy.org/, accessed on 15 February 2020) with an e-value cutoff of 1 × 10^−5^ [40]. Abundances of KEGG pathways, COG functional genes, and CAZymes were all normalized into counts per million reads (cpm) for downstream comparison analysis, with the cutoff of cpm >5 in at least 50% of animals within each group.

### 2.6. Statistical Analysis

The relative abundance of bacteria, archaea, viruses, and eukaryota, the gene of CAZymes at class level, and COG at category level among 4 studied groups were compared using the Kruskal–Wallis H test, with *p*-value < 0.05 being regarded as significantly different. Using the free online platform of Majorbio Cloud Platform (http://www.majorbio.com, accessed on 15 April 2020), linear discriminant analysis coupled with effect size (LEfSe) was performed to identify the microbial taxa differentially represented among groups at phyla to genera levels [41] with the cutoff of LDA score > 2.5 for bacteria or LDA score > 3 for archaea, and *p*-value < 0.05. The abundances of pathways KEGG, COG, and CAZy at the family level were compared among four groups using LEfSe with the cutoff of LDA score > 2 and *p*-value < 0.05. To explore the relative contribution of taxa to the rumen-enriched CAZymes and KEGG pathways, the taxonomic information for each selected gene (at genus level) was extracted, and their relationships were calculated using the nonparametric Kruskal–Wallis H test analysis of variance (*p* < 0.05) followed by multiple comparisons with Bonferroni correction, and then the similarity percentage analysis [42] was executed using PAST software [43] to rank taxa according to their contribution to differences of taxonomic profiles as described by Zhang et al., 2017 [44] and Ofek-Lalzar et al., 2014 [45].

### 2.7. Nucleotide Sequence Accession Numbers

The metagenomics raw sequences of rumen samples in this study were accessible from the NCBI Sequence Read Archive (SRA) with the accession number SRX10500085-SRX10500099.

## 3. Results

### 3.1. Profiling of the Rumen Metagenome

This metagenome sequencing experiment of 15 dairy buffalo rumen samples generated 132.21 GB of raw bases in total. After filtering the host genome data, a total of 129.25 GB of clean bases were obtained for all samples. A total of 11,546,554 contigs with lengths from 300 to 415,681 bp were obtained (Appendix A). A total of 26,965,974 ORFs and 17,545,624 nonredundant genes were predicted with an average length of 475.47 and 528.46 bp, respectively.

In the present study, taxonomic classification was performed using DIAMOND software (version 0.8.35) to map the sequence of nonredundant gene catalog against the NR database with an e-value cutoff of 1 × 10^−5^. A total of 5 domains, 8 kingdoms, 137 phyla, 297 classes, 695 orders, 1,303 families, and 3,256 genera were obtained. At the domain level, bacteria (96.59 ± 0.45%, mean  ±  standard error of the mean (SEM)) and archaea (2.29 ± 0.48%) were the most highly abundant taxa, followed by Eukaryota (0.31 ± 0.24%), viruses (0.28 ± 0.13%) and unclassified taxa (0.25 ± 0.01%) (Appendix A). Using the Kruskal–Wallis H test, we found that the relative abundance of bacteria, archaea, and viruses was significantly different among the four tested groups (*p* < 0.05; Table 1), while eukaryota and unclassified taxa were not different (*p* > 0.05). In addition, the dominant viral genera, including unclassified_f__Myoviridae (25.09 ± 3.52%, proportion of total virus) and unclassified_f__Siphoviridae (20.04  ± 3.17%, proportion of total virus), were all unclassified in our study (Appendix A). Thus, the downstream comparison of rumen microbial taxa among the four groups of dairy buffaloes was only focused on bacteria and archaea.

### 3.2. Comparison of Buffalo Rumen Microbiome across Different Age Groups

The dominant bacterial phyla included Bacteroidetes (51.25 ± 6.37%) and Firmicutes (29.73 ± 6.27%), and the dominant bacterial genera were *Prevotella* (35.19  ±  6.85%), followed by *unclassified_d__Bacteria* (9.14  ±  2.53%) and *Bacteroides* (7.87  ±  1.24%). For differential abundance comparison analysis, bacteria with LDA scores greater than 2.5 and *p* < 0.05 were regarded to have a different abundance across the four dairy buffalo groups. Our data showed that 6 clades were more enrichment in the Y group, 18 clades were more enrichment in the M group, 5 clades were more enrichment in the E group, and 17 clades were more enrichment in the O group (Figure 1). For the phylum level, significant enrichment of the relative abundance of Actinobacteria, Tenericutes, and Chlamydiae were characteristics of M buffaloes, while that of Fibrobacteres was characteristic of the O group (LDA > 2.5, *p* < 0.05). At the genus level, we identified significant enrichment of the relative abundance of *Lactobacillus* and *Phascolarctobacterium* were characteristics of the Y group (LDA > 2.5, *p* < 0.05), which could distinguish Y from the other three groups. Significant enrichment of the relative abundance of *Bifidobacterium*, *Mycoplasma*, *Chlamydia*, *Bacillus,* and *Burkholderia* were characteristics of the M group (LDA > 2.5, *p* < 0.05). Significant enrichment of the relative abundance of genera *Stenotrophomonas* and *Pullulanibacillus* were characteristics of the E group (LDA > 2.5, *p* < 0.05). While significant enrichment, the relative abundance of genera *Bacteroides*, *Paludibacter*, *Parabacteroides*, *Fibrobacter,* and *unclassified_p__Bacteroidetes* were characteristics of the O group (LDA > 2.5, *p* < 0.05). The characteristic bacterial genera with the largest enrichment in Y, M, E, and O groups were *Lactobacillus*, *Bifidobacterium*, *Stenotrophomonas,* and *Bacteroides*, respectively (LDA > 2.5, *p* < 0.05; Appendix A).

For the differential abundance comparison analysis of archaea, the relative abundance of *Methanobrevibacter*, the most abundant archaeal genus (80.33 ± 1.97%, 92.25 ± 0.65%, 86.16 ± 2.03 and 89.82 ± 0.85% for the Y, M, E, and O group, respectively), was significantly higher in the rumen of M dairy buffaloes than other 3 groups with Y had the lowest content of *Methanobrevibacter* (LDA > 3, *p* < 0.01), while the abundances of other differential genera, including *Methanobacterium* (Y: 3.09 ± 0.55%; M: 1.05  ± 0.05%; E: 1.24  ± 0.08%; O: 1.16  ± 0.13%), *Methanosarcina* (Y: 1.99 ± 0.24%; M: 0.94 ± 0.11%; E: 1.71  ± 0.24%; O: 1.18 ± 0.04%) and *Methanothermobacter* (Y: 1.06  ± 0.27%; M: 0.11  ± 0.01%; E: 0.15  ± 0.04%; O: 0.16  ± 0.04%) were all significantly higher in the rumen of Y dairy buffaloes than other 3 groups (LDA > 3, *p* < 0.01; Figure 2 and Appendix A).

### 3.3. Changing Pattern of Rumen Microbial Functions with Increasing Age

#### 3.3.1. CAZyme Functional Annotation

In the present study, a total of 515 genes encoding class-level CAZymes were identified. Glycoside hydrolases (GHs, 51.60 ± 1.10%) genes were the most highly represented, followed by genes encoding glycosyltransferase (GTs, 19.26 ± 0.78%), carbohydrate esterases (CEs, 15.06 ± 0.58%), and carbohydrate-binding modules (CBMs, 10.23 ± 0.84%); a few genes associated with polysaccharide lyases (PLs, 2.61 ± 0.28%) and encoding auxiliary activities (AAs, 1.15 ± 0.08%) (Appendix A). At the family level, GT2 was predominant, followed by CE1, GH3, GT4, CE10, GH2, GH31, GH109, GH97, GH28, GT51, GH13, and GH10 (Appendix A). At the class level, among the four groups with different ages, the proportion of genes encoding GHs and CEs decreased significantly with age, and buffaloes in group Y had the greatest GHs and CEs, while the proportion of genes encoding CBMs and PLs increased significantly with age and the buffaloes in group O had the greatest CBMs and PLs genes, compared with other three groups (*p* < 0.05; Figure 3A). At the family level, LEfSe analysis (LDA > 2) showed that among the genes encoding CAZymes that deconstruct cellulose, hemicellulose, starch, protein, and lignin, 22 CAZymes were enriched in the Y group, including 17 GHs, 2 CEs, 2 CBMs, and 1 AA, with the GH3 was dominant; 9 differential CAZymes were enriched in M group, including 4 GTs, 3 GHs, and 2 CBMs; GT2 and GT4 were enriched in the rumen of E group; 24 differential CAZymes were enriched in the rumen of O group, including 10 GHs, 5 GTs, 4 CBMs, 4 PLs, and CE11, with the CBM37 was the dominant (LDA > 2, *p* < 0.05; Figure 3B).

#### 3.3.2. COG Functional Annotation

A cluster of orthologous groups of proteins (COG) annotation for 5,735,175 nonredundant genes was performed using Diamond (version 0.8.35) [37] based on the eggNOG database with an e-value cutoff of 1 × 10^−5^, and a total of 36,418 COG functional genes were obtained. Most of the genes were ascribed to function unknown (S), transcription (K), cell wall/membrane/envelope biogenesis (M), carbohydrate transport and metabolism (G), and replication, recombination, and repair (L) in the COG database (Appendix A). The proportion of genes in the category of metabolism decreased with age (*p* < 0.05), while the proportion of genes in the category of poorly characterized increased with age (*p* < 0.01). In terms of the proportion of genes classified in the categories of cellular processes and signaling, information storage and processing increased with age, but no significant differences were observed among the four groups (Figure 3C). LEfSe analysis showed that the number of genes at the function level involved in G (carbohydrate transport and metabolism), P (inorganic ion transport and metabolism), E (amino acid transport and metabolism), and F (nucleotide transport and metabolism) were higher in Y when compared to the other three groups, while S (function unknown) and N (cell motility) were more abundant in the O group, Q (secondary metabolites biosynthesis, transport, and catabolism) was more abundant in the M group (LDA > 2, *p* < 0.05; Figure 3D).

#### 3.3.3. KEGG Functional Annotation

The KEGG annotation was conducted using Diamond [37] against the Kyoto Encyclopedia of Genes and Genomes database (https://www.genome.jp/kegg/, accessed on 15 February 2020) with an e-value cutoff of 1 × 10^−5^. In this study, a total of 5,735,175 nonredundant genes were annotated to 10,242 pathways. In the first-level category, the dominant pathways (average relative abundance of >5% for at least one group) included “Metabolism” (68.47 ±  0.93%), “Genetic information processing” (13.32 ± 0.53%), and “Environment information processing” (5.34 ± 0.36%). At the second level, the “Carbohydrate metabolism” (16.51 ± 0.33%), “Global and overview maps” (11.04 ± 0.18%), “Amino acid metabolism” (9.89 ± 0.23%), “Nucleotide metabolism” (7.50 ± 0.16%), “Replication and repair” (6.55 ± 0.33%) and “Energy metabolism” (6.06 ± 0.08%) being the most abundant (average relative abundance of >5% for at least one group).

When the identified KEGG pathways were compared at the third level, three pathways belonging to the metabolism of carbohydrates (starch and sucrose metabolism, butanoate metabolism, propanoate metabolism), three pathways related to the metabolism of energy (nitrogen metabolism, oxidative phosphorylation, carbon fixation pathways in prokaryotes), two pathways related to the metabolism of amino acids (alanine, aspartate, and glutamate metabolism, cyanoamino acid metabolism), two pathways belonging to environmental information processing of membrane transport (phosphotransferase system) and signal transduction (two-component system), the pathway belonging to the metabolism of xenobiotics biodegradation (drug metabolism-other enzymes) and the pathway belonging to the biosynthesis of other secondary metabolites (phenylpropanoid biosynthesis) were all more abundant in Y than other three groups (LDA > 2, *p* < 0.05; Figure 3E). Furthermore, the pathway of the citrate cycle belonging to the metabolism of carbohydrates, three pathways related to the metabolism of cofactors and vitamins (biotin metabolism, folate biosynthesis, nicotinate, and nicotinamide metabolism), two pathways belonging to the genetic information processing of folding, sorting and degradation (RNA degradation, protein export), the pathway related to the metabolism of terpenoids and polyketides (polyketide sugar unit biosynthesis), the biosynthesis of other secondary metabolites (acarbose and validamycin biosynthesis), the cellular processes of cell growth and death (cell cycle-caulobacter), environmental information processing of membrane transport (bacterial secretion system) and the organismal systems of aging (longevity regulating pathway-worm) were all significantly enriched in the rumen microbiomes of O group’s buffaloes (LDA > 2, *p* < 0.05; Figure 3E). In addition, four pathways (DNA replication, homologous recombination, mismatch repair, and base excision repair) belong to the genetic information processing of replication and repair, the pathway related to the genetic information processing of folding, sorting, and degradation (sulfur relay system), and the pathway belonging to the genetic information processing of translation (ribosome biogenesis in eukaryotes) were all significantly enriched in the rumen of M group’s buffaloes (LDA > 2, *p* < 0.05; Figure 3E).

### 3.4. Functional Contribution Analysis

To visualize the association between rumen-enriched taxonomic and functional properties, we determined the taxonomic origin of rumen-enriched functional attributes for all samples. The top 15 rumen-enriched CAZymes at the family level (including GT2, CE1, GH3, GT4, CE10, GH2, GH31, GH109, GH97, GH28, GT51, GH13, GH10, CBM50, and CE6) and the top 15 identified KEGG pathways (including biosynthesis of amino acids, purine metabolism, carbon metabolism, pyrimidine metabolism, starch and sucrose metabolism, aminoacyl-tRNA biosynthesis, ABC transporters, amino sugar, and nucleotide sugar metabolism, alanine, aspartate, and glutamate metabolism, glycolysis/gluconeogenesis, quorum sensing, homologous recombination, pyruvate metabolism, mismatch repair, and carbon fixation pathways in prokaryotes) were conducted for the taxonomic genus and functional contribution analysis. We found that *Prevotella* was the main contributor to those functions, but no significant difference was observed in the relative contribution of *Prevotella* to the KEGG (33.88 ± 1.55% of the normalized total relative contribution for the 15 functional categories, 31.70 ± 5.84%, 36.38 ± 4.51% and 30.37 ± 4.93% for the Y, M, E, and O group, respectively) and CAZyme (45.84 ± 1.42%, 43.31 ± 4.97%, 47.15 ± 4.26% and 38.52 ± 6.61% for the Y, M, E, and O groups, respectively) functional categories among four groups (paired *t*-test, *p* < 0.05). For the KEGG pathways at level 3, the relative contribution of the age-affected genus *Lactobacillus* to the 15 functional categories for the rumen of the Y group’s samples (4.32 ± 0.79% of the normalized total relative contribution for the 15 functional categories) was extremely significantly higher than M (0.11 ± 0.013%), E (0.13 ± 0.034%) and O (0.09 ± 0.016%) group (paired *t*-test, *p* < 0.01) (Figure 4A; Appendix A). Additionally, with increasing age, a reduced contribution by the age-affected genus Lactobacillus for the CAZyme at the family level was also observed (accounting for 1.74 ± 0.31% of the normalized total relative contribution for the rumen of Y samples, 0.31 ± 0.04% for the M samples, 0.24 ± 0.02% for the E samples and 0.23 ± 0.03% for the O samples, paired t-test, *p* < 0.01) (Figure 4B; Appendix A).

## 4. Discussion

The main objective of this experiment was to study the patterns and functions of rumen microorganisms of dairy buffaloes of different ages using metagenomic sequencing. Buffalo milk (BM) is ranked second after cow milk in the world as the BM produced is more than 12% of the world’s milk production [46]. It is a well-documented fact that age affects the total solids, solids-not-fat, lactose, and ash of BM. Furthermore, Xue et al. [1] proved that the rumen microbiome contributes to high-quality milk production. Previous research also confirmed that rumen bacterial colonization begins at birth and is extremely important for the physiological development of fully functional rumen capable of anaerobic fermentation [47,48]. Therefore, we have collected 15 rumen samples of dairy buffaloes with different physiological development stages of age from juvenile (12-month-old) to old (>9-year-old), namely 1-year-old buffaloes (Y group, parity = 0, n = 5), 3–5-year-old buffaloes (M group, parity = 1, n = 4), 6–8-year-old buffaloes (E group, parity = 2, n = 3) and >9-year-old buffaloes (O group, parity ≥ 3, n = 3), to exploring rumen microbial function in dairy buffaloes. Our metagenomic analysis of different age dairy buffalo rumen microorganisms confirmed that the bacteria were the most highly represented microorganisms (>95%). Notably, the relative abundance of archaea content in the rumen of dairy buffaloes observed in our study was higher than that reported in an earlier study of Surti buffalo (*Bubalus bubalis*) [25]. This may be attributed to factors such as differences in methodology, geographical locations, environmental adaptation, and different feedstuff.

Rumen microbial community composition varies by the animal developmental stage [4]. However, the microbial population dynamics from juvenile to older buffaloes are poorly understood, as only a limited study has been reported on ruminal bacterial diversity in buffaloes from birth to 1 year of old using 16S rRNA gene amplicon sequencing [47]. In the present study, we found that the dominant abundance of bacterial phyla (Bacteroidetes, Firmicutes, and Proteobacteria) and genera (*Prevotella* and *Bacteroides*) were in accordance with other research groups for buffaloes [23,26,47,49]. Bacteroidetes are the Gram-negative anaerobic bacteria, and previous studies showed that some members of the Bacteroidetes phylum are mainly responsible for carbohydrate degradation [50], whereas those belonging to the phylum Firmicutes play an important role in energy utilization [51,52]. Notably, the abundance of genus *Lactobacillus* was significantly higher in the young dairy buffaloes (Y) compared with all other three groups. Studies have demonstrated that *Lactobacillus* regulates gastrointestinal immune function and promotes animal health as a candidate bacterial therapeutics that mitigates the bovine respiratory pathogens [53], the promising cholesterol-lowering properties [54], anti-pathogens effect [55], and prevention of human and animal disease [56]. The significantly higher relative abundance of this genus in the rumen of young dairy buffaloes (Y) implies that the rumen of young dairy buffaloes has more probiotic functions than the rumen of older dairy buffaloes. Moreover, we also found that the abundance of genus *Prevotella* in the Y group was higher than in the other three groups, while no statistically significant difference was observed among the four groups (*p* > 0.05). Additionally, functional contribution analysis revealed that the genera *Prevotella* and *Lactobacillus* largely contributed to the starch and sucrose metabolism, carbon metabolism, and their roles in GT2, GH3, GT4, CE10, GH2, GH31, and GH109 production (Appendix A). All of those pathways and enzymes have the ability to metabolize starch and sucrose, hemicellulase and cellulase activities, and participate in the dairy buffalo rumen microbial carbohydrate metabolism, thereby affecting the 12-month-old dairy buffaloes (Y) rumen carbohydrate metabolism. We could speculate that the rumen environment of aged dairy buffaloes may not be suitable for the survival and growth of the genera *Prevotella* and *Lactobacillus*. The lower relative abundance of *Prevotella* and *Lactobacillus* in the rumen of the O group might cause decreased immunity and weakened ability to resist disease in older age animals; therefore, the rumen bacteria in older dairy buffaloes have a lower breakdown capacity of complex carbohydrates than that in younger dairy buffaloes. It should be pointed out that the far more abundant *Prevotella* are likely to be more responsible for polysaccharide degradation than the less abundant *Lactobacillus*, despite the fact that the latter is elevated in the Y group (*p* < 0.05). The variation of *Lactobacillus* might be more responsible for the host’s immunity and the ability to resist disease. However, this assumption requires further investigation.

Ruminal archaea constitute 0.3–3.3% of rumen microbes and are primarily responsible for rumen methane production [57,58]. Moreover, the feature of methane emission is related to the phylum Euryarchaeota, including the orders such as Methanobacteriales, Methanomicrobiales, and Methanosarcinales. Among them, *Methanobrevibacter* is the most commonly encountered genus within Methanobacteriales [59]. In the present study, the genus *Methanobrevibacter* was most abundant (86.71 ± 5.60%), followed by *Methanobacterium* (1.79  ± 1.17%), which is in agreement with other ruminant studies conducted in Indian buffalo (*Bubalus bubalis*) [60] or another study in which *Methanobrevibacter* accounted for the majority of methanogens in the rumen [59]. However, a previous study has also confirmed that the genus *Methanomicrobium* was the most dominant component of methanogen populations in Murrah buffaloes (*Bubalus bubalis*) in North India [61]. The reasons for those different results might be ascribed to the sample collection/extraction and processing method differences in methodology and adaptations of buffaloes to different dietary conditions in different parts of the world. Furthermore, our research results showed that the proportion of *Methanobrevibacter* content in the M group was significantly higher than in the other three groups, with the Y group having the lowest content of *Methanobrevibacter* (*p* < 0.05, LDA > 3). The Methanobrevibacter affiliated to the phylum Euryarchaeota, the only known microorganisms capable of methane production, is the main rumen methanogens [62]. Overall, younger animals had slightly more diverse methanogenic populations than older buffaloes. Further studies are necessary to determine if these differences have any impact on methane emissions. Furthermore, previous studies have reported lower diversity and variability in archaeal communities compared to other microbes in the rumen with the change of species or diets [63], and there were several reports investigating differences in bacterial community diversity and richness [64]. However, such comparable data on rumen methanogenic communities for dairy buffaloes are lacking. Possible reasons are due to either the low abundances of rumen archaea (3–4% of total microbes) or the establishment of stable methanogenic communities in the rumen at a very early age [65]. Therefore, more studies are required in the future to determine their abundance and their contribution to the host metabolism.

The well-documented fact is that age affected the total solids, solids-not-fat, lactose, and ash of dairy buffaloes [46]. The age effect on microbial community and the physiological response from the animal host is rarely reported in ruminants, and to the best of our knowledge, no relevant studies on the dairy buffaloes. Previous studies of rumen microbial communities in calves (<2 years old) reported the existence of age-related changes in the rumen microbiota [4,66]. However, the average culling time for dairy buffaloes worldwide is generally longer than that of dairy cows (>10 vs. 5~6 years old), indicating that these existing data are not sufficiently comprehensive. Age-related effects could result in hosts developing a specific model of microbiota and cooperating with them to produce effects and metabolism peculiar to the host, such as reported for the family Christensenellaceae in the human gut [67]. At the same time, age and physiological status were also reported to have effects on shifts in the rumen microbiome [5,57,68]. According to a report, differences in methane production across a wide age range (9–10 months of heifers, 45–65 months of young adults, and 96–120 months of older adults) in populations of dairy cows are mainly influenced by developmental physiological changes [9]. We, therefore, could speculate that rumen microbiota respond or adapt to host age; this suggests that the taxa composition and functions may differ in host populations of different aged dairy buffaloes. In the present study, the functional annotation based on the COG database showed that the nonredundant genes in the four groups were mainly related to the energy and metabolism necessary for the survival of rumen microbes. We found that the number of genes involved in carbohydrate transport and metabolism and amino acid transport and metabolism were significantly higher in Y as compared to the other three groups, the number of genes involved in function unknown, and cell motility were significant lower in Y as compared to the other three groups (LDA > 2, *p* < 0.05). It is suggested that there is a more active energy and material interaction between the young dairy buffaloes and their microbes. Cell motility is associated with biomembrane components. It is reported that the aging cell is characterized by function changes in cell membrane permeability, reduced metabolism, and substance transport function [69]. However, no studies have demonstrated age-related cell types in ruminant hosts, which requires further research in the future. For the KEGG analysis, we found that the starch and sucrose metabolic and two-component system were significantly higher in the Y as compared to the other three groups, while the homologous recombination, mismatch repair, DNA replication, nucleotide excision repair, and RNA degradation were significantly higher in the O as compared to the other three groups. The bacterial two-component regulatory system plays an important role in bacteria’s perception of changes in the external environment and the regulation of its own gene expression [70]. A previous study has shown that with the gradual increase in age, the accumulation of genetic damage will accelerate aging, which may lead to more active regulatory pathways such as gene repair in aged animals [71]. Therefore, the results of this study showed that the normal healthy activities of the dairy buffaloes at different ages are different. We thus speculated that the rumen microbial function may have some adaptive changes with age. In addition to this, pyruvate metabolism and oxidative phosphorylation were enriched in rumen microorganisms of 12-month-old dairy buffaloes as compared with rumen microbes of over 9-year-old dairy buffaloes. Propionic acid metabolism is related to stimuli for rumen development and the maturation process [72] and the rumen microorganisms that degrade cellulose [73], and up-regulation of oxidative phosphorylation pathway can enhance the activity of microbial cells [74]. In summary, more functions related to protein metabolism and fewer functions belonging to carbohydrate metabolism were enriched in the rumen of older dairy buffaloes. Previous studies have shown that excessive protein fermentation often leads to the production of toxic chemical substances such as NH_3_, H_2_S, and phenols [75], while the lack of carbohydrate metabolism-related genes may decrease the potential to generate beneficial compounds, such as short chain fatty acids (SCFA), which can protect the intestinal tract from damage. It is well known that the rumen epithelium acts as a protective barrier between the rumen and the host, but it can be damaged by toxic compounds, acidosis, or a high-concentrate diet feeding can impair rumen epithelium function [76,77]. Thus, we speculate that younger dairy buffaloes might have a lower risk of rumen damage. However, this assumption requires further investigation.

A sum of 515 different enzyme classes was found. Among them, 246 different glycoside hydrolases (GHs) that hydrolyze the glycosidic bond among the carbohydrate molecules were identified. Most of the identified GHs were polysaccharide-degrading enzymes such as cellulase, hemicellulose, and pectinase, all of which degrade the plant cell wall to produce oligosaccharides [78]. More complete degradation of feed polysaccharides will result in greater total VFA production, which will increase nutrient delivery to the host [79]. In our study, among the four groups of different ages, the proportion of genes encoding GHs and CEs decreased significantly with age, while the proportion of genes encoding CBMs and PLs increased significantly with age (*p* <  0.05). Lots of significantly enriched GHs identified in the rumen of Y, suggesting the high feed nutrients degradation and utilization efficiency in the rumen of younger dairy buffaloes than in the other three groups. The most highly represented GH family was GH3 (which includes β-glucosidase, xylan 1,4-β-xylosidase, β-glucosylceramidase, β-N-acetylhexosaminidase, and β-L-arabinofuranosidase), followed by GH2, which comprises β-galactosidase, β-mannosidase, β-glucuronidase, α-L-arabinofuranosidase, mannosylglycoprotein endo-beta-mannosidase, exo-beta-glucosaminidase [80], GH31, GH109, GH97, GH28, GH13, GH4, and GH51, and *Prevotella* were the major organism from which the source of those enzymes (Appendix A). Our results are consistent with the previous findings on the rumen of Indian buffalo, which reported that GH3 and GH2 were also the predominant GHs [22]. In accordance with previous research results, genera such as *Prevotella*, *Bacteroides*, *Faecalibacterium*, *Clostridium*, *Butyrivibrio,* and *Ruminococcus* are commonly reported for their role in GHs production were relatively abundant in Y group animals compared to O group buffaloes [81]. Moreover, we also found CE1 (acetyl xylan esterase) and CE10 (arylesterase), which act on side chains of hemicellulose and pectin, respectively, and render both the large molecules accessible for further breakdown [82], in the rumen of Y were also significantly higher than other three groups (*p* < 0.05), while GT4 (mainly sucrose synthase activity), GH28 (polygalacturonase activity) and GH13 (α-amylase activity) of Y group were significantly lower than the other three groups (*p* < 0.05). Those findings suggest that there exist some certain differences in the rumen’s degradation ability of carbohydrate substrates of dairy buffaloes of different ages. These results are in line with our previous research on dairy buffaloes [28]. On the basis of the aforementioned experimental results, we speculate that the reasons for the variation among different groups of those enzymes may be that genus *Lactobacillus* and *Prevotella**,* which largely contributed to the pathways of starch and sucrose metabolism, carbon metabolism, and for their role of GHs production, are more abundant in the rumen of Y group’s dairy buffaloes, whereas the rumen environment of older age dairy buffaloes may not be suitable for the survival of the *Lactobacillus* and *Prevotella* [22]. Collectively, significant enrichment of genes encoding CAZymes that break down the carbohydrates (GHs and CEs) in the rumen of the Y group provides clues that juvenile dairy buffaloes were might more capable of deconstructing complex substrates than old buffaloes.

The limitation of our study lies in the small sample size selected for the experimental dairy buffaloes with the nearly same parity and numbers in each group due to providing adequate background information on the studied dairy buffaloes. However, the sample size in our study was at least three per group, which basically meet the requirements of statistical analysis, and our results are valuable to know the dynamic changes of dairy buffaloes’ rumen microbiota with different age and provide insights into strategies for manipulating dairy buffaloes’ rumen microbiota for animal maturity and health. The rumen is a huge biological resource fermenter that contains numerous enzymes that degrade plant fibers, but the understanding of the rumen enzymes is still relatively lacking. Therefore, like a previous study [83], based on our metagenomics research results, we could more easily identify some novel putative bacterial glycoside hydrolases and carbohydrate esterases from the rumen of older dairy buffaloes, while the polysaccharide lyases can be more easily obtained in the rumen of younger dairy buffaloes. Moreover, microbial transplantation has been proposed as one of the promising methods of reshaping the symbiotic microbiota to treat rumen disorders in ruminants or to enhance productivity and health [84,85,86]. Thus, younger dairy buffaloes are a potential candidate for the donor rumen microbial transplantation for older dairy buffaloes or cattle for improvements in both animal welfare and profitability. Moreover, the probiotics found in this study are also informative in regulating the growth and health of calves.

## 5. Conclusions

This study describes the patterns and functions of the rumen microbiome of dairy buffaloes of different ages, and the results demonstrate that the Bacteroidetes and Firmicutes were the predominant bacteria, with *Prevotella* accounting for the highest abundance at the genus level of the rumen microbiome of dairy buffaloes. *Methanobrevibacter* was constituted the most dominant archaeal genera across all samples and showed the lowest relative abundance of this genus in the rumen of Y. Further studies are necessary to determine if these differences have any impact on methane emissions. The characteristic bacterial genera with the largest abundance in the Y, M, E, and O were *Lactobacillus*, *Bifidobacterium*, *Stenotrophomonas,* and *Bacteroides*, respectively. The genus *Lactobacillus* largely contributed to the pathways of starch and sucrose metabolism, carbon metabolism and for their roles in GT2, CE1, GH3, GT4, CE10, GH2, GH31, and GH109 production, thus affecting the 12-month-old dairy buffalo rumen carbohydrate metabolism. Most differed COG and KEGG pathways were enriched in Y with carbohydrate metabolism, while older dairy buffaloes enriched more functions of protein metabolism and the processing of replication and repair, indicating that younger dairy buffaloes might have a lower risk of rumen damage. The results expanded our understanding of the dynamic changes of dairy buffaloes’ rumen microbiota from juvenile to old age and provided insights into strategies for manipulating buffaloes’ rumen microbiota.

## Figures and Tables

**Figure 1 microorganisms-10-01491-f001:**
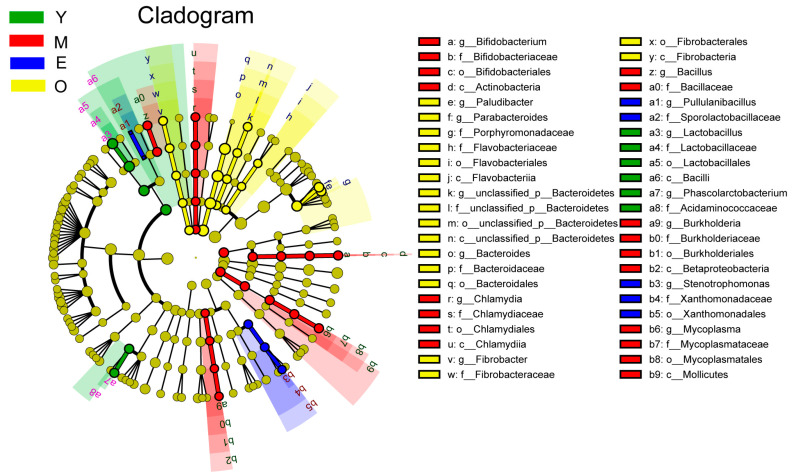
The dairy buffalo rumen bacteria showing different abundance values among four groups (highlighted by small circles and by shading). There are five layers from the inside of this plot to the outside, corresponding to five levels of taxonomy (phylum, class, order, family, and genus). The nodes (small circle) with different colors represent bacterial characteristics of the corresponding groups with a higher relative abundance compared with that in the other three groups, while yellow nodes indicate the bacteria that are not statistically and biologically differentially abundant among the four groups. Significant differences were confirmed by linear discriminant analysis effect size (LEfSe) analysis with the cutoff of LDA > 2.5 and *p* < 0.05.

**Figure 2 microorganisms-10-01491-f002:**
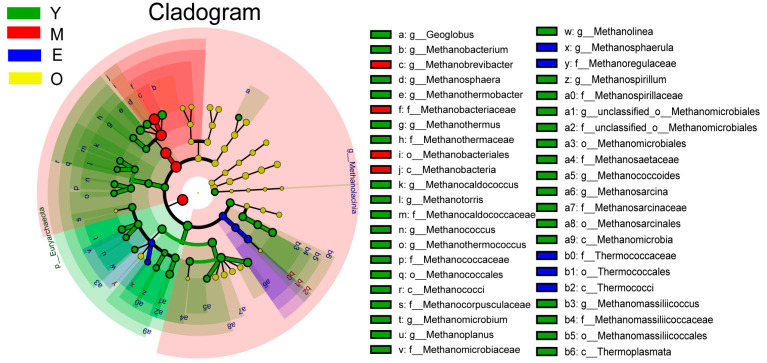
The dairy buffalo rumen archaea showing different abundance values among four groups (highlighted by small circles and by shading). There are five layers from the inside of this plot to the outside, corresponding to five levels of taxonomy (phylum, class, order, family, and genus). The nodes (small circle) with different colors represent archaeal characteristics of the corresponding groups with the higher abundance compared with that in the other three groups, while yellow nodes indicate the bacteria that are not statistically and biologically differentially abundant among the four groups. Significant differences were confirmed by linear discriminant analysis effect size (LEfSe) analysis with the cutoff of LDA > 3 and *p* < 0.05.

**Figure 3 microorganisms-10-01491-f003:**
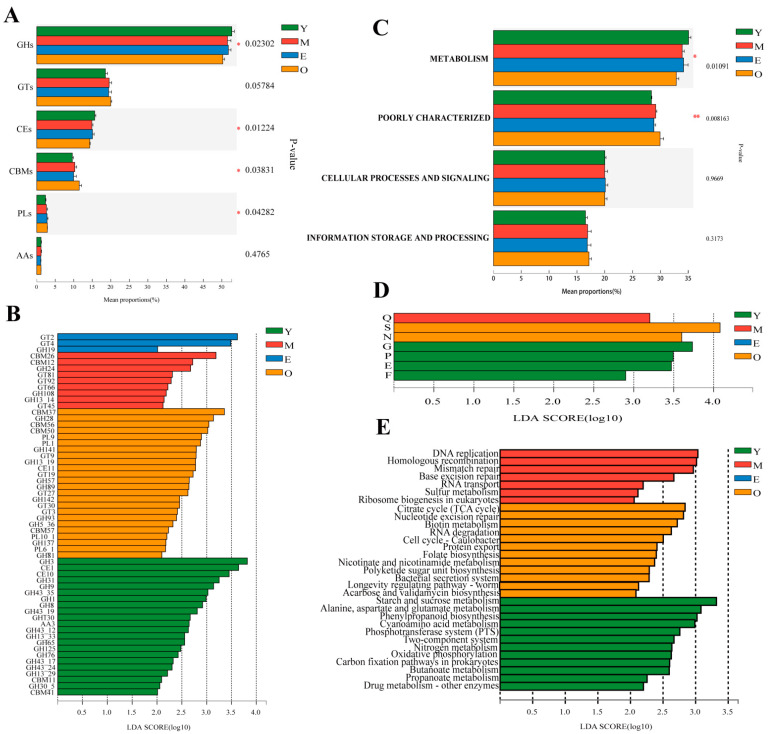
Functional difference of ruminal microbiome of dairy buffaloes. Comparison of predicated CAZyme genes of dairy buffalo rumen microbiota at class (**A**) and family (**B**) levels. Difference analysis of predicted COG genes of dairy buffalo rumen microbiota at category (**C**) and function (**D**) levels. (**E**) Histogram of the LDA scores computed for differentially abundant predicted genes of dairy buffalo rumen microbiota at the KEGG pathway level 3 classification. Significant differences were confirmed by linear discriminant analysis effect size (LEfSe) analysis with the cutoff of LDA > 2 and *p* < 0.05. GH: glycoside hydrolases; GT: glycosyltransferase; CE: carbohydrate esterases; CBM: carbohydrate-binding modules; PL: polysaccharide lyases; AA: auxiliary activities; *, *p* < 0.05; **, *p* < 0.01.

**Figure 4 microorganisms-10-01491-f004:**
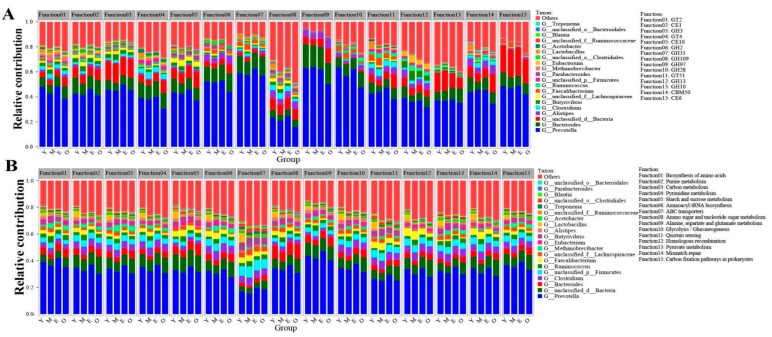
Relative contribution of different taxa (genus levels) to identified rumen-enriched functional attributes of CAZymes encoded genes at the family level (**A**) and KEGG pathway level 3 (**B**) in different samples. GH: glycoside hydrolases; GT: glycosyltransferase; CE: carbohydrate esterases; CBM: carbohydrate-binding modules.

**Table 1 microorganisms-10-01491-t001:** Comparison of dairy buffalo rumen microbiota at the domain level.

Domain	Groups ^1^				SEM	*p*-Value
	Y	M	E	O		
Bacteroidetes	97.80 ^a^	95.38 ^b^	97.54 ^a^	95.97 ^b^	0.302	0.013
Archaea	1.18 ^a^	3.13 ^b^	1.46 ^a^	2.81 ^b^	0.262	0.010
Eukaryota	0.63	0.84	0.42	0.64	0.107	0.783
Viruses	0.13 ^a^	0.41 ^b^	0.32 ^b^	0.33 ^b^	0.034	0.009
unclassified	0.26	0.24	0.26	0.25	0.004	0.306

^1^ Y: 1-year-old group, M: 3–5-year-old group, E: 6–8-year-old group, and O: > 9-year-old group. Comparison of the percentage relative abundance of microbial domains in dairy buffalo rumens. ^a,b^ Within rows in the upper-right corner differ (*p* < 0.05) if without a common letter.

## Data Availability

The data sets generated for this study can be found in NCBI Sequence Read Archive, SRX10500085-SRX10500099.

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
