# Peer review of "An Age Effect of Rumen Microbiome in Dairy Buffaloes Revealed by Metagenomics"

_microorganisms, 2022, doi:10.3390/microorganisms10081491_

Round 1
Reviewer 1 Report
In this study, the authors applied a shotgun metagenomic sequencing to rumen samples from dairy buffalos to provide comprehensive metagenomic fluctuation by age. Although the number of samples used in this paper is small (3-5 animals per group) and needs to be added as a limitation of this paper, the authors have conducted a comprehensive analysis of the microbiota and functional analysis to identify characteristics that could be affected by age. This study could potentially give new insights into metagenomic data of dairy buffalo rumens: however, the authors still need some corrections. Below please find my comments.
Comments:
1. Limitations in this study, such as the small sample size used in this study, should be included in the discussion. If possible, please also add a discussion of how the results of this study will be applied to practical husbandry.
2. Materials and Methods; Even if it is the same as previously reported, the diet and rearing environment could have a significant impact on the microbiota and should be briefly described in the paper.
3. Please add the version to the MEGAHIT and CD-HIT tools you used, and any other tools whose versions are not yet described.
4. Line 108; Please delete one of the duplicate reference numbers.
5. Line 190 and other parts; I am not sure there is sufficient evidence to conclude that anything above a certain threshold in the LEfSe analysis is a biomarker. Were these phylum groups not detected at all in the other groups? There is no discussion of relative abundance or cutoff values in each group. As mentioned above, there is also the problem of the small sample size of each group used in this study. Therefore, I would suggest that the wording be modified to "were characteristic of" or similar.
6. Line 231; Not “family-level” but “class-level”?
Author Response
We thank the reviewer for reading our manuscript and for giving useful comments, and giving us the chance to revise the manuscript. Please see the attachment for the point-by-point response to the comments.

Reviewer 2 Report
The authors performed a very comprehensive metagenomic study on the ruminal microbial community of dairy buffaloes. The methodologies for obtaining the metagenomic data apper to be sound, and the data themselves greatly extend our knowledge of the ruminal community in dairy buffaloes, a species of considerable economic importance. However, the authors’ interpretation of the metagenomic data leaves the reviewer largely uncomfortable. Many of the conclusions drawn by the authors seem simplistic and not well coupled to the animals themselves. Statements are made regarding methane production, or efficiency of polysaccharide digestion, that are pure speculation, and are not backed up by any measurements of these variables in the animals. In fact, there are so many intervening steps between a gene and a physiological response, that it is risky to try to link these so tightly in the absence of other measurements.
Specific comments:
Figure 1: The data in this table would work much better in a Table format, due to the massive domination by the Bacteria making the other domains nearly invisible in the Figure.
L189-198: The reviewer disputes that the indicator taxa are “biomarkers”. While the particular taxa may be significantly more abundant in a particular age group, the taxa are not diagnostic of the group.
L367-368: Could it also perhaps be due to differences in methods of sampling, DNA isolation and sequencing?
L375-378: This sentence is confusing. Firstly, Bacteroidetes are not unique in their capacity to degrade high molecular weight organic matter, nor are they limited to this particular role. Secondly, what is meant by the statement that Firmicutes “mainly be responsible for energy utilization”? All bacteria generate energy in their catabolism, and use this energy in their anabolism. How do Firmicutes differ from other bacteria in this regard?
L394-398: This is a hard sell. While the authors present data that Lactobacillus encodes certain CAZymes, these are but a small fraction of the total from all the bacteria, and Lactobacillus species have never been shown to be significantly cellulolytic or hemicellulolytic. By contrast, some other genera (Prevotella, Ruminococcus) that are far more abundant, are known to actively degrade cellulose or hemicellulose. The decrease in Lactobacillus abundance with host age from ~5% to well under 1%, while dramatic, is unlikely to have much impact on degradation of the types of fiber typically fed to buffaloes.
L403-405: The two stated abundances together exceed 100%.
L413-415: What is the basis for this speculation? Are the authors claiming that younger buffaloes produce more methane because they have a higher proportion of methanogens that were not in the genus Methanobrevibacter? Why would this be? Don’t each of these groups produce similar methane yields per unit substrate converted? And wouldn’t methane emissions be governed less by relative abundance and more by absolute abundance of methanogens (which was not measured), or by the amount of H2 supplied by the fermentative bacteria?
L450-453: The authors’ point is not clear. Are they discussing the aging of bacterial cells, or the aging of the host? The two are not directly coupled: There is no evidence of which the reviewer is aware that older ruminant hosts have older bacterial cells than do younger hosts.
L481-483: Do the authors mean instead that polysaccharide-degrading enzymes (rather than oligosaccharide-degrading enzymes) produce oligosaccharides?
L483-485: It is not clear how the abundance of oligosaccharide-degrading enzymes will affect the “efficiency’ of VFA production (whatever that means), unless these enzymes were rate-limiting in the VFA production pathway. Can the authors cite a literature source to support this claim?
L491-495: It is interesting that only a few of the enzymes mentioned here are active on the major polysaccharides in dairy feeds (cellulose, xylans, pectin).
L514-517: This statement provides a testable hypothesis regarding a superior ability to degrade feed polysaccharides in younger buffaloes, but it does not provide any “evidence”. Is there, in fact, any literature that shows that “juvenile buffaloes are more capable of deconstructing complex substrates than old buffaloes”?
L522-525: As per comment to L413-415 above, this statement is not supported, and certainly should not be one of the main conclusions of this study.
L535-536: The second half of the sentence is wildly speculative. There is nothing in the data that provides strategies for ruminal manipulation, particularly as attempts at ruminal manipulation has proven to be almost universally unsuccessful.
Minor edits:
L159: Delete either “This” or “The”.
L185: Change “phyla” to “phylum”.
L257: Change ‘was decrease’ to “decreased”.
L334: Change “effected” to “affected”. Also L339.
L372: Change “limited study have been reported that” to “a limited study has been reported on”.
L384: Change “implying” to “implies”.
L406: Change “other” to “another”.
L407: Change “have” to “has”.
L441: Change comma to semicolon.
Author Response
We thank the reviewer for reading our manuscript and for giving useful comments, and giving us the chance to revise the manuscript. We have carefully checked the conclusions and statements well coupled to the studied animals, and provide some more evidence or references to back up of our conclusions. We are also apologize for that some of our interpretations of the metagenomic data have caused some misunderstandings and discomfort to the reviewer. We have made changes as reviewer suggested in the revised manuscript. Thank you. Please see the attachment for the point-by-point response to the comments

Reviewer 3 Report
An age effect of rumen microbiome in dairy buffaloes revealed by metagenomics
This article attempts to enlighten the reader on the impact age has on the rumen microbiome of dairy buffaloes. Using a study group of 15 animals (3x1yo, 3x3-5yo,3x6-8yo, and 3x9+yo). The study found that Bacteroides and Methanobrevibacter were significantly increased with age and goes on to describe patterns and functions of rumen microbiome of dairy buffaloes. Methanobrevibacter was the dominant archaeal genera across all samples, and showed the lowest relative abundance in the rumen 1yo dairy buffalo. The abundant bacteria observed were Lactobacillus, Bifidobacterium, Stenotrophomonas and Bacteroides, respectively. The genus Lactobacillus was largely contributed the pathways for Starch and sucrose metabolism, Carbon metabolism and for their roles of GT2, CE1, GH3, GT4, CE10, GH2, GH31 and GH109 production. The COG and KEGG pathways for carbohydrate metabolism were enriched in the 1yo buffalo, while older dairy buffaloes enriched more functions of protein metabolism and the processing of replication and repair. The authors hope that the results from this study could go towards expanding the understanding of the dynamic changes of the rumen microbiome in dairy buffaloes from juvenile to old age.
Broad comments.
This article is well organised and attempts to provide the basis data for further metagenomic studies on buffalo. The metagenomic sequencing data has been evaluated in a wide range of analysis, and appears well thought through. Due to the low number of studies in buffalo so far, studies into the changes over a number of years are highly interesting. There are however a number of small points that need to be addressed:
1. The introduction, although well rounded on most aspects, would benefit from expanding further on how the rumen of buffalo differs from other dairy herds, including a few more references, as this would help to add weight to the study.
2. Line 54-57: The author should avoid the use of highly subjective wording (“stronger digestive capacity” and “superiority”, without clearly defining what “stronger” means and how superiority would be judged.
3. More detail is required for the materials and methods section in relation to the sampling used. As the methods used for sampling and processing can have a huge impact on the outcome of the study it is essential that this information is abundantly clear to the reader.
4. Please insert the relevant reference for the CTAB method used for DNA extraction.
5. Please provide the analytical data (tabulated DNA quality/concentrations and gel images) on DNA quality to prove that the DNA used was high molecular weight and clean of contaminants.
6. In the results section (Line 176-178), the authors recommend the focus to be on only bacteria and archaea due to there abundance in the observed sequences. This will however, bias the study away from some potentially important findings, some evaluation of the eukaryotic/virus and unclassified could possibly show something interesting and as such for the author to completely ignore this seems an odd choice.
7. Throughout the entire article no mention has been made on the compositional nature of the data collected for this metagenomic study. Importantly this should really be incorporated into the data analysis, but at the very least this needs to addressed in the discussion.
8. Line 365-368 the author attempts to compare against a previous study but neglects to mention the study method (16S). Although a comparison can be made between methods it is also widely shown that different methods in analysis / sample collecting and processing can all impact the results of the study.
9. Line 421-422 the author gives possible reasons for differences to previous studies, but again fails to mention any sample collection/extraction method of analytical differences that could also account for these observed differences.
10. Line 477-478 the author speculates as to rumen damage but has given no evidence that rumen damage actually occurs, please clarify this statement.
11. Line 509-517 the author makes a number of speculative statements without any references to back up the thought process behind the ideas.
Author Response

(The authors gave the same response as above.)

Round 2
Reviewer 2 Report
The authors have improved the manuscript and have provided responses to reviewer comments that provide clarify some of the more confusing statements in the original manuscript. On reading these, it appears that some of the problem lies in the authors’ imprecise use of language that tends to obfuscate what they are trying to say. The reviewer thus makes a few additional suggestions:
L203: Be more specific in the legend, as there is no mention of measurement units. “Comparison of the percentage relative abundance of microbial domains in dairy buffalo rumens”.
L216-217: While the change from “biomarkers” to “were characteristic of” represents an improvement, it is still somewhat problematic, as it implies these taxa were present in one group, but not in another. The reviewer suggests the authors be more specific, by stating “significant enrichment the relative abundance of (name taxa) was characteristic of (name group).” The inclusion of “enrichment” better describes the basis of comparison, and that despite significant differences, all taxa were present in all groups.
L417-419: The authors have improved the text in this section, but there are a couple of remaining issues. First, they state, “Moreover, we also found that the abundance of genus Prevotella in the Y group was higher than other three groups, while no significant difference was observed among four groups.” This statement is confusing, as the first part states that abundance in the Y group was higher than in the other three, but the second part states that there were no differences. Thus, clarification is needed. Second, the reviewer still believes that the authors need to emphasize that the far more abundant Prevotella are likely to be more responsible for polysaccharide degradation than are the less abundant Lactobacillus, despite the fact that the latter are elevated in the Y group.
L445-464: Despite the authors’ helpful addition of several qualifying statements, it is still not clear what underlies their hypothesis that the Y group “seems to present greater potential for produce less methane compared to older dairy buffaloes.” Again, the authors do not explain what underlies this hypothesis. Is it just because the relative abundance of Methanobrevibacter is lower in the Y group? Because the relative abundance of other methanogens is higher in this group, it would seem that this would cancel out this effect, even if total methanogen abundance was directly correlated with total methanogen abundance. The reviewer suggests dropping this hypothesis altogether, and stating that further studies are necessary to determine if these differences have any impact on methane emissions.
L526-529: Here again, the meaning is not clear. The cited reference [79] makes no mention of “efficiency” of VFA production. The reviewer suggests restating their point to emphasize that more complete degradation of feed polysaccharides will result in greater total VFA production, which will increase nutrient delivery to the host. This statement is not seriously debatable, and gets around any confusion regarding “efficiency”.
Author Response
Response to Reviewer 2 Comments (2)
The authors have improved the manuscript and have provided responses to reviewer comments that provide clarify some of the more confusing statements in the original manuscript. On reading these, it appears that some of the problem lies in the authors’ imprecise use of language that tends to obfuscate what they are trying to say. The reviewer thus makes a few additional suggestions:
Response: We thank the reviewer for reading our revised manuscript and for giving useful additional suggestions, and giving us the chance to revise the manuscript.
L203: Be more specific in the legend, as there is no mention of measurement units. “Comparison of the percentage relative abundance of microbial domains in dairy buffalo rumens”.
Response: Thanks the reviewer, we have made the changes in the legend as suggested in the revised manuscript.
L216-217: While the change from “biomarkers” to “were characteristic of” represents an improvement, it is still somewhat problematic, as it implies these taxa were present in one group, but not in another. The reviewer suggests the authors be more specific, by stating “significant enrichment the relative abundance of (name taxa) was characteristic of (name group).” The inclusion of “enrichment” better describes the basis of comparison, and that despite significant differences, all taxa were present in all groups.
Response: Thanks the reviewer’ suggests and we have made the changes as suggested in the revised manuscript as following:
Our data showed that six clades were more enrichment in the Y group, eighteen clades were more enrichment in the M group, five clades were more enrichment in the E group, and seventeen clades were more enrichment in the O group (Figure 1). For the phylum level, significant enrichment the relative abundance of Actinobacteria, Tenericutes and Chlamydiae were characteristics of M buffaloes, while that of Fibrobacteres was characteristic of O group (LDA>2.5, P<0.05). At the genus level, significant enrichment the relative abundance of Lactobacillus and Phascolarctobacterium were characteristics of Y group (LDA>2.5, P<0.05), which could distinguish Y from the other 3 groups. Significant enrichment the relative abundance of Bifidobacterium, Mycoplasma, Chlamydia, Bacillus and Burkholderia were characteristic of M group (LDA>2.5, P<0.05). Significant enrichment the relative abundance of genera Stenotrophomonas and Pullulanibacillus were characteristics of E group (LDA>2.5, P<0.05). While significant enrichment the relative abundance of genera Bacteroides, Paludibacter, Parabacteroides, Fibrobacter and unclassified_p__Bacteroidetes were characteristics of O group (LDA>2.5, P<0.05). The characteristic bacterial genera with the largest enrichment in Y, M, E and O groups were Lactobacillus, Bifidobacterium, Stenotrophomonas and Bacteroides, respectively (LDA>2.5, P<0.05; Figure S2A).
L417-419: The authors have improved the text in this section, but there are a couple of remaining issues. First, they state, “Moreover, we also found that the abundance of genus Prevotella in the Y group was higher than other three groups, while no significant difference was observed among four groups.” This statement is confusing, as the first part states that abundance in the Y group was higher than in the other three, but the second part states that there were no differences. Thus, clarification is needed. Second, the reviewer still believes that the authors need to emphasize that the far more abundant Prevotella are likely to be more responsible for polysaccharide degradation than are the less abundant Lactobacillus, despite the fact that the latter are elevated in the Y group.
Response: First, what we want to express is that although the abundance of genus Prevotella in the Y group was higher than other three groups, there is no statistically significant difference was observed among four groups (P>0.05). We agreed with the reviewer’s opinion that the far more abundant of Prevotella are likely to be more responsible for polysaccharide degradation than the less abundant of Lactobacillus, despite the fact that the latter are elevated in the Y group (P<0.05). The variation of Lactobacillus might be more responsible for host’s immunity and the ability of resist disease. We have made the changes as suggested in the revised manuscript as following:
Moreover, we also found that the abundance of genus Prevotella in the Y group was higher than other three groups, while no statistically significant difference was observed among four groups (P>0.05). ...... It should be pointed out that the far more abundant of Prevotella are likely to be more responsible for polysaccharide degradation than the less abundant of Lactobacillus, despite the fact that the latter are elevated in the Y group (P<0.05). The variation of Lactobacillus might be more responsible for host’s immunity and the ability of resist disease. However, this assumption requires further investigation.
L445-464: Despite the authors’ helpful addition of several qualifying statements, it is still not clear what underlies their hypothesis that the Y group “seems to present greater potential for produce less methane compared to older dairy buffaloes.” Again, the authors do not explain what underlies this hypothesis. Is it just because the relative abundance of Methanobrevibacter is lower in the Y group? Because the relative abundance of other methanogens is higher in this group, it would seem that this would cancel out this effect, even if total methanogen abundance was directly correlated with total methanogen abundance. The reviewer suggests dropping this hypothesis altogether, and stating that further studies are necessary to determine if these differences have any impact on methane emissions.
Response: Agreed, thanks. We have deleted this hypothesis, and rewrote as “further studies are necessary to determine if these differences have any impact on methane emissions.” in the revised manuscript. We have also made changes in the “Conclusions” section in the revised manuscript.
L526-529: Here again, the meaning is not clear. The cited reference [79] makes no mention of “efficiency” of VFA production. The reviewer suggests restating their point to emphasize that more complete degradation of feed polysaccharides will result in greater total VFA production, which will increase nutrient delivery to the host. This statement is not seriously debatable, and gets around any confusion regarding “efficiency”.
Response: We thank the reviewer for this advice. We have made the changes as suggested in the revised manuscript as following:
More complete degradation of feed polysaccharides will result in greater total VFA production, which will increase nutrient delivery to the host [79].
A point-by-point response to the reviewer’s comments can also be found the attachment.
